# Air-Stable and Highly Active Transition Metal Phosphide Catalysts for Reductive Molecular Transformations

Takato Mitsudome [1,2,3]

1  Department of Materials Engineering Science, Graduate School of Engineering Science, Osaka University, 1-3 Machikaneyama, Toyonaka 560-8531, Osaka, Japan; mitsudom@cheng.es.osaka-u.ac.jp
2  Innovative Catalysis Science Division, Institute for Open and Transdisciplinary Research Initiatives (ICS-OTRI), Osaka University, Suita 565-0871, Osaka, Japan
3  Precursory Research for Embryonic Science and Technology (PRESTO), Japan Science and Technology Agency (JST), 4-1-8 Honcho, Kawaguchi 333-0012, Saitama, Japan

**Abstract:** This review introduces transition metal phosphide nanoparticle catalysts as highly efficient and reusable heterogeneous catalysts for various reductive molecular transformations. These transformations include the hydrogenation of nitriles to primary amines, reductive amination of carbonyl compounds, and biomass conversion, specifically, the aqueous hydrogenation reaction of mono- and disaccharides to sugar alcohols. Unlike traditional air-unstable non-precious metal catalysts, these are stable in air, eliminating the need for strict anaerobic conditions or pre-reduction. Moreover, when combined with supports, metal phosphides exhibit significantly enhanced activity, demonstrating high activity, selectivity, and durability in these hydrogenation reactions.

**Keywords:** metal phosphide; nanoparticle; hydrogenation; heterogeneous catalyst; green chemistry

## 1. Introduction

### 1.1. Challenges and Innovations in Non-Precious Metal Catalysis for Hydrogenations

Catalytic hydrogenation using molecular hydrogen ($H_2$) is a vitally important and atom-efficient method in modern chemical production, suitable for a wide range of applications ranging from milligram-scale organic synthesis to multi-ton-scale chemical production. In the industrial hydrogenation, traditionally, sponge metal catalysts based on non-precious metals such as nickel and cobalt, which have a high surface area, have been prevalently employed in these hydrogenation reactions [1,2]. However, these sponge metals, epitomized by Raney catalysts, suffer from significant drawbacks [3–5]. Their instability in the air, leading to rapid oxidation and deactivation, poses a major challenge. The highly pyrophoric nature of these sponge metals necessitates handling under the exclusion of air, limiting their applicability. Furthermore, the low catalytic activities of sponge metals require harsh reaction conditions such as high temperature and high $H_2$ pressure. Therefore, these drawbacks have led to a growing demand for the development of more stable and active non-precious metal catalysts.

Recent advances have witnessed a surge in reports detailing the development of non-precious metal nanoparticle catalysts, which demonstrate higher activity compared to traditional sponge metal catalysts [6–8]. Despite these advancements, the air instability issue of non-precious metal nanoparticle catalysts remains unresolved. These novel catalysts frequently require the in situ generation of nanoparticles using $H_2$ reduction at elevated temperatures prior to their application. The air instability not only raises safety concerns due to ignition risks, but also impedes the precise design of nanoparticle catalysts, including aspects such as morphology control and composite formation with a support material. Additionally, it complicates the structural analysis of the actual active species. Consequently, the development of air-stable non-precious metal catalysts would not only enhance the safety profile of reaction systems, but also expand the possibilities for precise

design of catalysts and analytical evaluation. Such progress could potentially lead to the discovery of even more active catalysts. In this context, our group has embarked on developing non-precious metal catalysts that are both air-stable and highly active. Our efforts have led to the discovery that nanosized transition metal phosphides function as non-precious metal catalysts, combining atmospheric safety, high activity, and durability for a wide range of hydrogenation reactions including the hydrogenation of nitriles to primary amines, reductive amination of carbonyl compounds, and the aqueous hydrogenation reaction of mono- and disaccharides to sugar alcohols.

*1.2. Advancements in Transition Metal Phosphide Catalysts*

The study of transition metal phosphide catalysis began with Sweeny's report in 1958 [9], demonstrating that nickel phosphide catalysts were effective in the vapor-phase hydrogenation of nitrobenzene. Research expanded throughout the 1970s and 1980s to include the hydrogenation of alkenes, dienes, and alkynes [10–14], dehydrogenation of methylcyclohexane [15], and the dimerization of isobutylene [14] using primarily nickel phosphide. From 1990 onwards, transition metal phosphides were recognized for their high activity and durability in hydrodenitrogenation [16] and hydrodesulfurization [17–25], both of which are crucial reactions in petroleum refineries. The 2010s saw a surge in studies on transition metal phosphides as electrocatalysts for the hydrogen evolution reaction [26–30]. The pioneering work has been achieved by the R. E. Schaak. They demonstrated that shape-controlled $Ni_2P$ nanoparticles were the most active non-precious metal-based electrocatalysts for this reaction at the time [31]. Subsequently, other nanosized non-precious metal phosphides including CoP [32,33], FeP [34], MoP [35,36], Co–Fe–P [37], and Ni–Co–P [38] have also been reported as effective catalysts for this reaction. Currently, the catalytic performance and structure–activity relationship of these catalysts in hydrogen evolution reaction represents one of the most vibrant areas of research in transition metal phosphide catalysis. A comprehensive bibliometric analysis using Scifinder, with criteria focusing on the inclusion of "phosphide" and "catalyst" in the title, abstract, or keywords, revealed a substantial body of literature comprising 6008 publications. The year 2023 marked a peak with 831 publications, underscoring the growing research interest in this field.

In addition to their established roles in hydrodenitrogenation, hydrodesulfurization, and hydrogen evolution reaction, recent years have seen a burgeoning interest in the application of non-precious metal phosphides to liquid-phase molecular transformations. This emerging area of research encompasses various reactions, including the semihydrogenation of alkynes [39–41], hydrogenation of polar functional groups [42–45], and coupling reactions [46,47]. Despite these advances, the full potential of non-precious metal phosphides in catalyzing a broader range of liquid-phase molecular transformations remains largely untapped. This gap in research offers a fertile area for future studies aiming to discover new sustainable pathways for chemical synthesis utilizing transition metal catalysts. This review highlights the advancements in cobalt and nickel phosphide nanoparticle catalysts, with a focus on their performance in hydrogenation reactions. By investigating the underlying factors that contribute to their catalytic efficiency, we facilitate future innovations in green and sustainable chemistry. The exploration of such catalysts across diverse molecular transformations not only deepens our understanding of their catalytic mechanisms, but also introduces new opportunities for environmentally friendly chemical processes. To this end, Figure 1 illustrates these themes in a conceptual diagram, offering an at-a-glance overview of the synthesis, characterization, and application of air-stable and highly active transition metal phosphide catalysts in reductive molecular transformations.

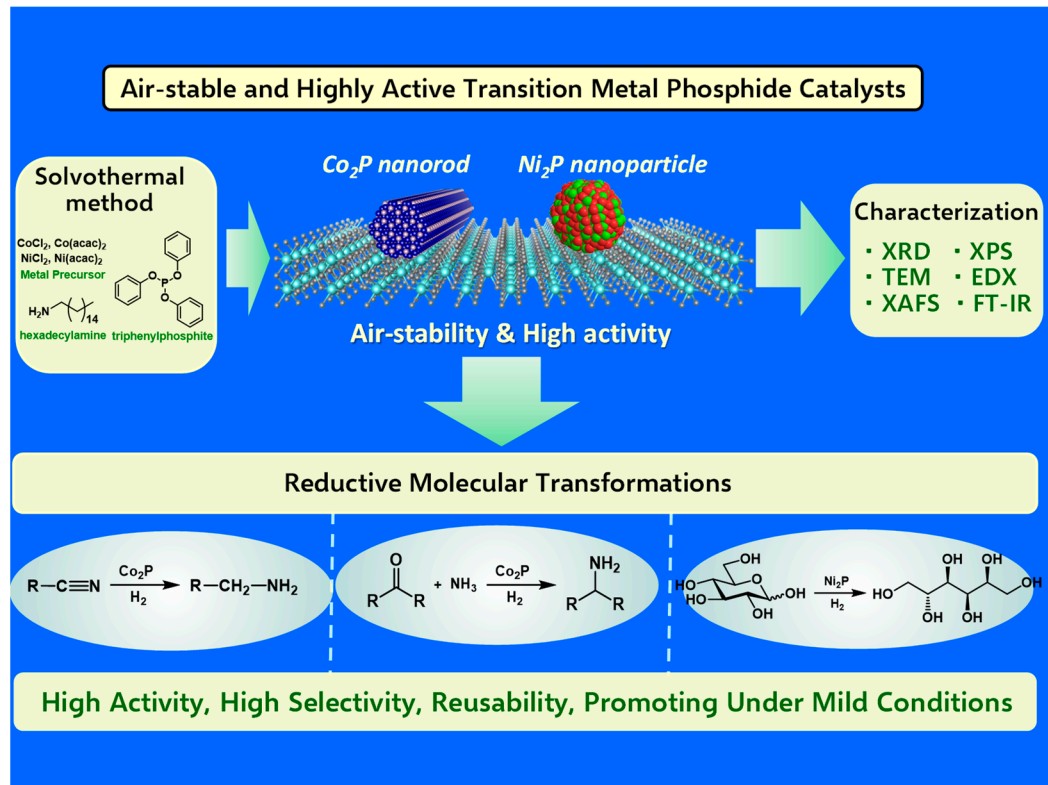

**Figure 1.** Conceptual Diagram: Overview of Air-Stable and Highly Active Transition Metal Phosphide Catalysts Synthesis, Characterization, and Applications in Reductive Molecular Transformations.

## 2. Catalytic Performance of Metal Phosphide Nanoparticle Catalysts

### 2.1. Cobalt Phosphide Nanoparticle Catalysts

Cobalt phosphide nanoparticle is prepared via the solvothermal method. Cobalt phosphides with an hexagonal-columnar structure with 20 nm in length and 9 nm in width (nano-$Co_2P$) can be obtained by the thermal decomposition of cobalt chloride and triphenyl phosphite in 1-octadecene in the presence of hexadecylamine, as depicted in Figure 2a,b [48,49]. The composition of the cobalt phosphide nanorods was confirmed as $Co_2P$ through selected area electron diffraction (SAED) patterns (Figure 2c), X-ray diffraction (XRD) patterns, and elemental analysis by energy dispersive X-ray spectroscopy (EDS) (Figure 2d–f). The prepared nano-$Co_2P$ is non-pyrophoric and can be handled in air. Furthermore, the nano-$Co_2P$ is easily immobilized on various metal-oxide supports under atmospheric conditions by redispersing it in hexane followed by the addition of the supports.

#### 2.1.1. Hydrogenation of Nitriles Using Nano-$Co_2P$ Catalysts

The hydrogenation of nitriles is one of the most important methods for the synthesis of primary amines, which are widely used as raw materials and intermediates in pharmaceuticals, agrochemicals, polymers, surfactants, and dyes. Theoretically, nitrile hydrogenation is an ideal reaction with 100% atomic efficiency as it does not produce any by-products. Traditionally, the hydrogenation of nitriles in the industry has relied on cobalt and nickel-based catalysts, such as Raney or sponge metal catalysts, due to their cost-effectiveness [50,51]. However, these catalysts often require high catalyst loadings and operate under severe reaction conditions, including high hydrogen pressures ranging from 200 bar to 400 bar [51,52]. To address these limitations, some metal complex catalysts based on iron [53–55], cobalt [56–58], and manganese [59–61] have been developed. These catalysts promote the nitrile hydrogenation at lower hydrogen pressures of 30 bar to 60 bar. Despite these advancements, homogeneous catalysts present challenges in terms of catalyst

recovery and reuse, as well as potential contamination risks from dissolved metals. In contrast, recent studies have introduced stable, non-precious metal-based heterogeneous catalysts for nitrile hydrogenation [62–66]. Among these, the Beller group has reported stable cobalt nanoparticles derived from metal-nitrogen complexes [62,63] and metal–organic framework [64]. These catalysts facilitate nitrile hydrogenation under even lower hydrogen pressures of 2.5 bar to 30 bar. Although these nanostructured materials offer advantages as reusable non-precious metal catalysts, they still necessitate pressurized hydrogen and exhibit relatively low activity. Therefore, the pursuit of a highly active catalyst for efficiently transforming nitriles under mild conditions remains a critical area of research.

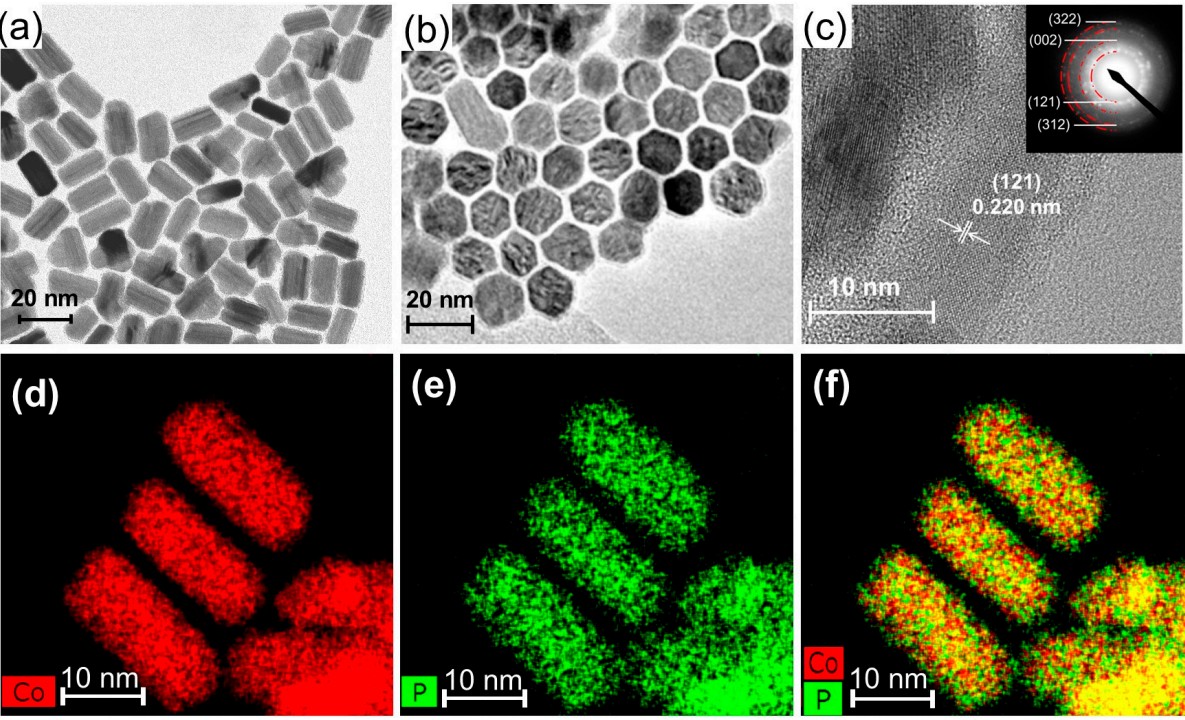

**Figure 2.** Characterization of nano-$Co_2P$. (**a**) Side view TEM image of nano-$Co_2P$ showing a nanorod morphology. (**b**) Top view TEM image of nano-$Co_2P$ showing the hexagonal phase structure. (**c**) HR-TEM image of nano-$Co_2P$ with SAED pattern (inset). Elemental mapping images of (**d**) Co and (**e**) P. (**f**) Composite overlay image of (**d**,**e**). Adapted from [48] Mitsudome, T.; Sheng, M.; Nakata, A.; Yamasaki, J.; Mizugaki, T.; Jitsukawa, K. A Cobalt Phosphide Catalyst for the Hydrogenation of Nitriles. *Chem. Sci.* **2020**, *11*, 6682–6689.

Upon investigating the catalysis of supported nano-$Co_2P$ for the nitrile hydrogenation, it was discovered that nano-$Co_2P$ supported on hydrotalcite (HT: $Mg_6Al_2(OH)_{16}CO_3 \cdot 4H_2O$) (nano-$Co_2P$/HT) is an extremely effective catalyst for the hydrogenation of nitriles to primary amines [48]. Nano-$Co_2P$/HT was able to hydrogenate various nitriles including aliphatic nitriles (Table 1, entries 1–4), aromatic nitriles (entries 5 and 6), heterocyclic nitriles (entries 7–10), and substituted aromatic nitriles (entries 11–15), providing the corresponding primary amines with high yields. Nano-$Co_2P$ is also effective for polynitriles containing multiple nitrile groups within their molecules, as demonstrated in Figure 3. For example, 1,6-hexamethylenediamine (**16b**), a significant precursor of Nylon-6,6, can be obtained from adiponitrile with a high yield. Similarly, the biomass derivative sebaconitrile is smoothly transformed into the corresponding 1,10-diaminodecane (**17b**). Furthermore, this catalyst is applicable to tetranitriles, enabling the synthesis of important amines (**20b**) used as intermediates in the production of functional materials such as dendrimers. Unlike previously reported non-precious metal catalysts that require high $H_2$ pressure to facilitate the reaction, nano-$Co_2P$/HT exhibits high activity under atmospheric $H_2$ pressure, selectively yielding primary amines (Figure 4).

**Figure 3.** Hydrogenation of multi-nitriles using nano-Co$_2$P catalyst. Reaction conditions: catalyst, substrate (0.5 mmol), 2-propanol, NH$_3$ aq. (1.2 mL). [a] NH$_3$ aq. (0.6 mL), 4 h. [b] 4 h. [c] 10 h. [d] Substrate (0.07 mmol), 100 °C, 50 bar H$_2$, 3 h.

**Table 1.** Hydrogenation of various nitriles using nano-Co$_2$P/HT.

| Entry | Substrate | | Time (h) | Product | | Yield (%) |
|---|---|---|---|---|---|---|
| 1 | | (1a) | 1 | | (1b) | 94 |
| 2 | | (2a) | 2 | | (2b) | 99 |
| 3 | | (3a) | 4 | | (3b) | 88 |
| 4 | | (4a) | 4 | | (4b) | 91 |
| 5 | | (5a) | 1 | | (5b) | 93 |
| 6 | | (6a) | 1 | | (6b) | 99 |
| 7 | | (7a) | 2 | | (7b) | 92 |
| 8 [a] | | (8a) | 5 | | (8b) | 92 |
| 9 | | (9a) | 4 | | (9b) | 95 |
| 10 | | (10a) | 2 | | (10b) | 93 |
| 11 | | (11a) | 1 | | (11b) | 93 |
| 12 | | (12a) | 2 | | (12b) | 92 |
| 13 | | (13a) | 2 | | (13b) | 90 |
| 14 | | (14a) | 2 | | (14b) | 94 |
| 15 [a] | | (15a) | 5 | | (15b) | 88 |

Reaction conditions: catalyst (0.1 g), substrate (0.5 mmol), 2-propanol (3 mL), Aq. NH$_3$ (1.2 mL), [a] Aq. NH$_3$ (0.4 mL).

This represents the first example of a non-precious metal catalyst promoting the hydrogenation of nitriles under atmospheric $H_2$ conditions.

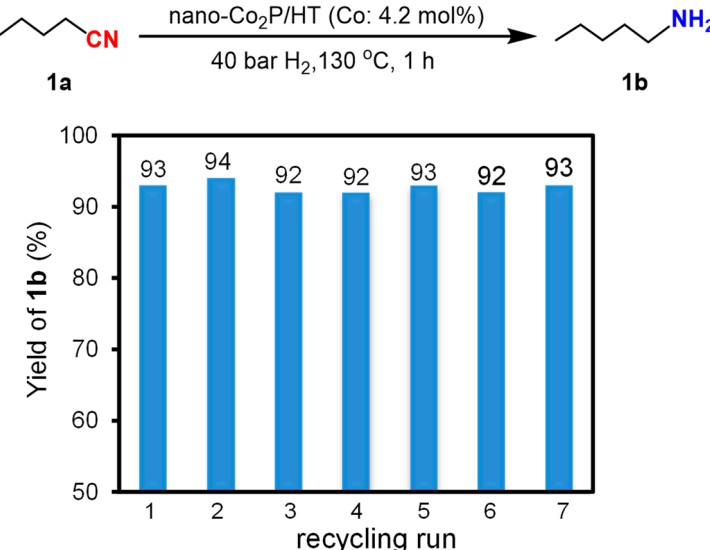

**Figure 4.** Hydrogenation of nitriles under ambient pressure of $H_2$ using nano-$Co_2P$. Reaction conditions: catalyst (0.1 g), substrate (0.5 mmol), 2-propanol (3 mL), $NH_3$ aq. (1.2 mL). [a] 130 °C. [b] $NH_3$ aq. (0.6 mL). [c] 16 h. [d] Catalyst (0.2 g), $NH_3$ aq. (0.6 mL).

After the reaction, nano-$Co_2P$/HT can be recovered from the reaction mixture through centrifugal separation under atmospheric conditions and can be reused without the need for any prereduction treatment. The result of the recycling experiment, as depicted in Figure 5, demonstrates that the yield of the reaction remains consistently high at $(93 \pm 1)\%$ across up to seven cycles. This air-stability and durability markedly contrast with previously reported non-precious metal catalysts, which tend to deactivate easily under atmospheric conditions, highlighting an outstanding feature of nano-$Co_2P$/HT.

**Figure 5.** Reuse experiments of nano-$Co_2P$/HT in hydrogenation of **1a** to **1b**.

2.1.2. Reductive Amination of Carbonyl Compounds Using Nano-$Co_2P$ Catalysts

One characteristic feature of nanoparticle synthesis via the solvothermal method is the production of nanoparticles with various morphology by altering the metal precursors and surfactants. In the synthesis of cobalt phosphide nanoparticles, switching the cobalt source from cobalt chloride to cobalt acetylacetonate results in rod-shaped nanostructures approximately 50–150 nm in length and about 10 nm in width (Figure 6a,b) [67,68]. The

high-resolution transmission electron microscopy (TEM) image of cobalt phosphide shown in Figure 6c reveals lattice spacings corresponding to the (020) and (113) planes of $Co_2P$, confirming that the prepared cobalt phosphide nanorods ($Co_2P$ NR) have the same $Co_2P$ composition as nano-$Co_2P$. $Co_2P$ NR stored at ambient conditions demonstrates high activity in the reductive amination of carbonyl compounds in aqueous media without the need for prereduction treatment with $H_2$ [15].

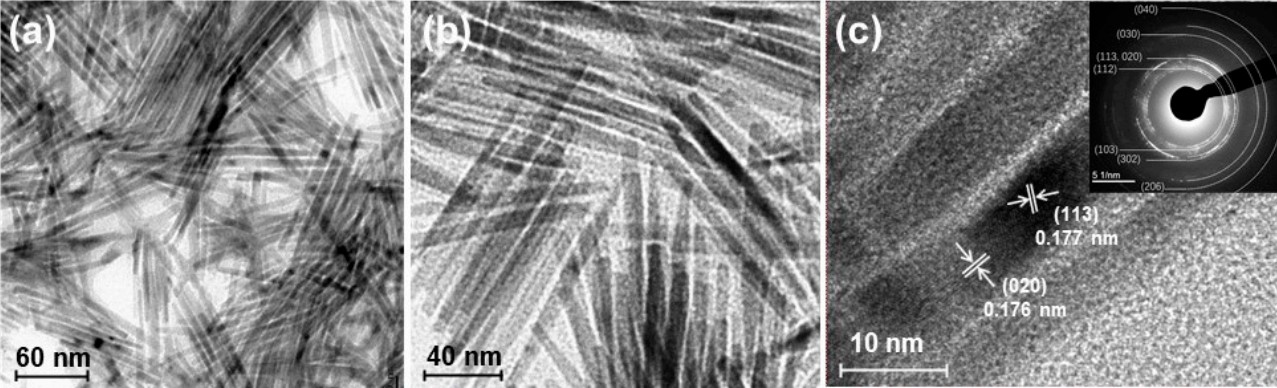

**Figure 6.** (**a**,**b**) TEM images of the $Co_2P$ NR showing a rod-like morphology. (**c**) HR-TEM image of the $Co_2P$ NR with the inset illustrating the corresponding SAED pattern. Adapted from [67] Sheng, M.; Fujita, F.; Yamaguchi, S.; Yamasaki, J.; Nakajima, K.; Yamazoe, S.; Mizugaki, T.; Mitsudome, T. Single-Crystal Cobalt Phosphide Nanorods as a High-Performance Catalyst for Reductive Amination of Carbonyl Compounds. *JACS Au* **2021**, *1*, 501–507.

The reductive amination of carbonyl compounds is an efficient and vital method for synthesizing primary amines [69–74]. This efficiency stems from the fact that carbonyl compounds are inexpensive and widely available starting materials. Furthermore, only water is theoretically formed as a by-product, making this process environmentally benign. Traditionally, reductive amination reactions in industrial processes have relied on air-sensitive and relatively low active nickel or cobalt sponge metal catalysts [75–77]. However, there have been significant advancements in enhancing the activity of heterogeneous catalysts based on earth-abundant metals [78–84]. Notably, recent progress has been made in the development of air-stable, non-precious metal catalysts. This advancement was achieved through the pyrolysis treatment of metal–nitrogen complex precursors, resulting in metal nanoparticles encapsulated within nitrogen-doped carbon layers [85–89]. These nickel, cobalt, and iron-based nanoparticles prepared by this method exhibit air stability and reusability in various fields of heterogeneous reactions, including the reductive amination reaction [90–93]. Nevertheless, these catalytic systems still require the use of flammable ammonia gas and/or high hydrogen pressures to facilitate the reaction. Additionally, the nitrogen-doped carbon layers can inadvertently shield the active sites on the catalyst surface, partially reducing their activity. Consequently, developing air-stable and highly active non-noble metal nanoparticle catalysts for reductive amination remains a significant challenge.

Table 2 presents the results of the reductive amination of benzaldehyde using the $Co_2P$ NR catalyst. Using an $NH_3$ aqueous solution or $NH_3$ gas as the aminating agent under 10 bar of $H_2$ at 100 °C, the target product, benzylamine, is obtained with high yield (entries 1–3). Moreover, $Co_2P$ NR can promote this reaction under atmospheric $H_2$ pressure (entry 4). In contrast, traditional catalysts such as sponge cobalt (sponge Co) and Co supported on $SiO_2$ followed by a prereduction treatment (Co/$SiO_2$-Red) show negligible progress in the amination reaction under atmospheric $H_2$ (entries 5 and 6). Additionally, using commercial bulk $Co_2P$ does not facilitate the reaction at all (entry 7). These results indicate that $Co_2P$ NR exhibits higher activity compared to conventional catalysts, and that the incorporation of phosphorus into cobalt and nanosizing are crucial

for enhancing cobalt's activity. Furthermore, $Co_2P$ NR demonstrates a broad range of substrate-applicability, efficiently converting various aldehydes and ketones into primary amines under atmospheric $H_2$ conditions, as illustrated in Figure 7.

**Table 2.** Reductive amination of benzaldehyde with $Co_2P$ NR and other cobalt catalysts.

| Entry | Catalyst | $H_2$ (bar) | NH$_3$ Source | Time (h) | Yield (%) | | |
|---|---|---|---|---|---|---|---|
| | | | | | 1d | 1e | 1f |
| 1 | $Co_2P$ NR | 10 | Aq. $NH_3$ | 10 | 93 | 0 | 0 |
| 2 [a] | $Co_2P$ NR | 10 | $NH_3$ gas | 10 | 88 | 0 | 0 |
| 3 [b] | $Co_2P$ NR | 10 | $NH_4OAc$ | 10 | 15 | 0 | 73 |
| 4 | $Co_2P$ NR | 1 | Aq. $NH_3$ | 12 | 90 | 0 | 1 |
| 5 | sponge Co | 1 | Aq. $NH_3$ | 12 | 0 | 11 | 3 |
| 6 | $Co/SiO_2$-Red | 1 | Aq. $NH_3$ | 12 | 0 | 0 | 0 |
| 7 | bulk $Co_2P$ | 1 | Aq. $NH_3$ | 12 | 0 | 0 | 0 |

Reaction conditions: Co catalyst (Co: 0.05 mmol), benzaldehyde (0.5 mmol), Aq. $NH_3$ 25% (3 mL), 100 °C. [a] $NH_3$ gas (2.5 bar), water (3 mL). [b] $NH_4OAc$ (0.2 g), water (3 mL).

**Figure 7.** Reductive amination of carbonyl compounds by $Co_2P$ NR at 1 bar $H_2$. Reaction conditions: $Co_2P$ NR (Co: 0.05 mmol), substrate (0.5 mmol), 12 h. [a] $NH_3$ aq. 25% (3 mL). [b] $NH_4OAc$ (0.1 g), ethanol (3 mL). [c] $NH_4OAc$ (0.15 g), ethanol (3 mL), 110 °C, 20 h.

2.1.3. Structure–Activity Relationship of Nano-$Co_2P$ Catalysts: Atmospheric Stability and Activity Factors

The results of X-ray absorption fine structure (XAFS) measurements of $Co_2P$ NR stored at room temperature under atmospheric conditions are presented in Figure 8. The K-edge X-ray absorption near edge structure (XANES) spectrum of $Co_2P$ NR shows that the

absorption edge energy is lower than that of CoO and similar to that of Co foil (Figure 8a). This suggests that the electronic state of Co in $Co_2P$ NR remains in a low oxidation state, close to metallic (zero-valent), even in the presence of air. This is further supported by XPS measurements of $Co_2P$ NR, where the binding energies of Co's $2p_{3/2}$ and $2p_{1/2}$ are 777.8 eV and 792.8 eV, respectively, closely aligning with those of metallic cobalt ($2p_{3/2}$ (777.9 eV) and $2p_{1/2}$ (793.5 eV)). Additionally, the Fourier transform (FT) plot of extended X-ray absorption fine structure (EXAFS) reveals that $Co_2P$ NR possesses metal–metal bonds (Co–Co bonds) around 2.3 Å, which are advantageous for $H_2$ activation (Figure 8b) [15].

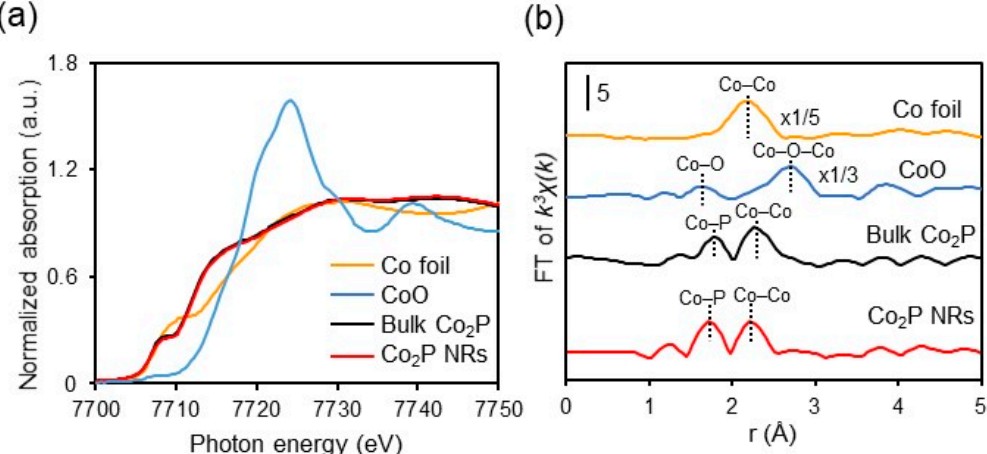

**Figure 8.** Co K-edge (**a**) XANES and (**b**) FT-EXAFS spectra of Co foil, CoO, bulk $Co_2P$, and $Co_2P$ NR. Adapted from [67] Sheng, M.; Fujita, F.; Yamaguchi, S.; Yamasaki, J.; Nakajima, K.; Yamazoe, S.; Mizugaki, T.; Mitsudome, T. Single-Crystal Cobalt Phosphide Nanorods as a High-Performance Catalyst for Reductive Amination of Carbonyl Compounds. *JACS Au* **2021**, *1*, 501–507.

Curve-fitting analysis further shows that, while the bulk $Co_2P$ has a coordination number ($CN_{Co–Co}$) of 4.0, $Co_2P$ NR exhibits a smaller $CN_{Co–Co}$ of 2.8, indicating that it contains a higher number of coordinatively unsaturated sites. These results reveal that the cobalt species in $Co_2P$ NR are stabilized in a low oxidation state and possess advantageous Co–Co bonds as well as a higher number of coordinatively unsaturated sites, contributing to its air-stability and high catalytic activity for hydrogenation.

To further investigate the air stability, structural changes of $Co_2P$ NR with increasing temperature in air were evaluated using XAFS measurements, as shown in Figure 9. The absorption edge energy gradually shifts to higher energies from 200 °C, with the intensity of the characteristic peak of oxides (7722 eV) increasing around 400 °C, indicating structural changes due to oxidation. Indeed, $Co_2P$ NR treated at 300 °C under atmospheric conditions exhibited significantly reduced catalytic activity compared to untreated $Co_2P$ NR, whereas $Co_2P$ NR treated at 200 °C maintained nearly equivalent catalytic activity. These findings robustly demonstrate that the active cobalt species in $Co_2P$ NR exhibit remarkable resistance against oxidative degradation.

### 2.2. Nickel Phosphide Nanoparticle Catalysts

Similar to the preparation method for cobalt phosphide nanoparticle, replacing cobalt with nickel results in the successful synthesis of nickel phosphide nanoparticles (nano-$Ni_2P$) with an average diameter of 5 nm and a composition of $Ni_2P$, as shown in Figure 10a. Energy-dispersive X-ray (EDX) line analysis in Figure 10b demonstrates the uniform dispersion of nickel and phosphorus within nano-$Ni_2P$ [94]. Additionally, nano-$Ni_2P$, such as cobalt phosphide, exhibits non-pyrophoricity and stability in air in a low oxidation state [95].

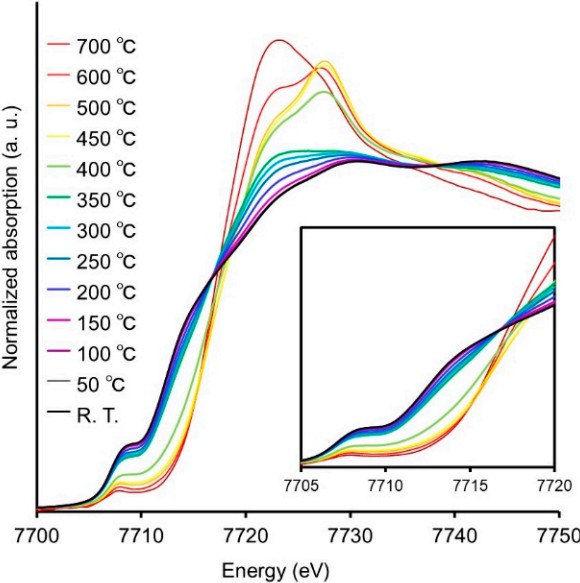

**Figure 9.** Co *K*-edge XANES spectra of Co foil, CoO, bulk Co$_2$P, and Co$_2$P NR.

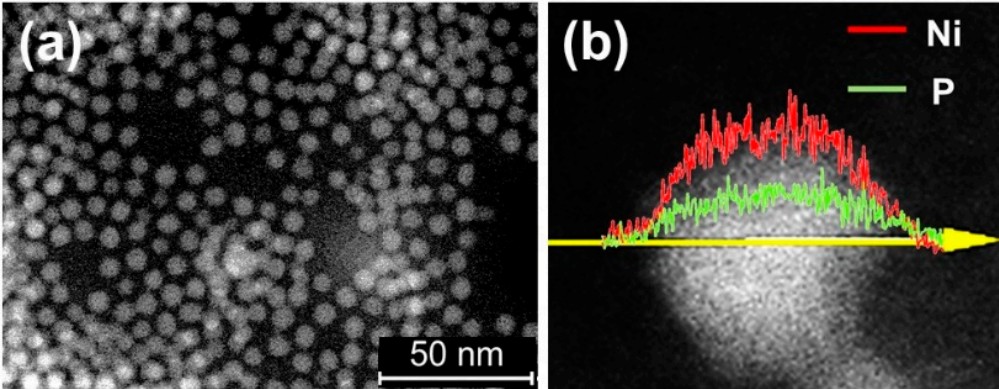

**Figure 10.** Characterization of nano-Ni$_2$P (**a**) TEM image of nano-Ni$_2$P and (**b**) EDX analysis of nano-Ni$_2$P. Adapted from [94] Fujita, S.; Nakajima, K.; Yamasaki, J.; Mizugaki, T.; Jitsukawa, K.; Mitsudome, T. Unique Catalysis of Nickel Phosphide Nanoparticles to Promote the Selective Transformation of Biofuranic Aldehydes into Diketones in Water. *ACS Catal.* **2020**, *10*, 4261–4267.

2.2.1. Synergistic Catalytic Effects with Support Materials

Traditional sponge metal catalysts, as well as supported non-noble metal nanoparticles, present handling difficulties due to their air sensitivity. This sensitivity significantly limits the scope for additional catalytic modifications of the metal species obtained. Furthermore, the selection of suitable supports for metal nanoparticles is confined to those that can endure high-temperature reduction treatment, a process necessary for nanoparticle formation. This constraint restricts the opportunity for creating composites of metal nanoparticles with various attractive support materials. Such a limitation, consequently, diminishes the possibilities for achieving improved catalytic performances. In contrast, nano-Ni$_2$P can be handled in air and allows for room-temperature compositing with suitable support materials for desired reactions [94–100].

Utilizing this feature, a nano-Ni$_2$P/HT composite can be prepared by combining nano-Ni$_2$P with hydrotalcite to decompose its layered structure near 250 °C. This nano-Ni$_2$P/HT catalyst exhibited high activity in the hydrogenation of carbonyl compounds in water, capable of reducing various carbonyl compounds under mild conditions [95]. Using acetophenone as a model substrate, the results of the hydrogenation reaction in aqueous solution under a 20 bar pressure of H$_2$ are shown in Table 3. With nano-Ni$_2$P,

the corresponding product 1-phenylethanol was obtained with a yield of 16%, while the use of nano-$Ni_2P$/HT increased the yield to 93%, significantly enhancing the activity (entry 1 vs. 2). Additionally, other oxide supports such as $Y_2O_3$, $ZrO_2$, $TiO_2$, $Al_2O_3$, and $Nb_2O_5$, excluding $SiO_2$, were effective, providing moderate-to-high yields (entries 4–9). Moreover, nano-$Ni_2P$/HT was active even under atmospheric $H_2$ pressure (entry 3). This high catalytic performance is attributed to the synergistically catalytic behavior where the oxygen-deficient sites of HT activate the carbonyl group of adsorbed acetophenone, which is then hydrogenated by $Ni_2P$ (discussed later).

**Table 3.** Hydrogenation of acetophenone using nano-$Ni_2P$ catalysts in water.

| Entry | Catalyst | Yield [%] |
|-------|----------|-----------|
| 1 | nano-$Ni_2P$ | 16 |
| 2 | nano-$Ni_2P$/HT | 93 |
| 4 | nano-$Ni_2P$/$Y_2O_3$ | 79 |
| 5 | nano-$Ni_2P$/$ZrO_2$ | 73 |
| 6 | nano-$Ni_2P$/$TiO_2$ | 63 |
| 7 | nano-$Ni_2P$/$Al_2O_3$ | 54 |
| 8 | nano-$Ni_2P$/$Nb_2O_5$ | 30 |
| 9 | nano-$Ni_2P$/$SiO_2$ | 7 |

Reaction conditions: catalyst (5 mol% metal), acetophenone (0.5 mmol), water (3 mL), $H_2$ (20 bar), 100 °C, 1 h. [a]$H_2$ (1 bar), 150 °C, 12 h.

### 2.2.2. Application of in Biomass Conversion

In molecular transformations aimed at the efficient utilization of biomass, it is desirable to develop solid catalysts based on non-noble metals that are low-cost, highly active, and durable for converting a large number of biomass-derived compounds into high-value-added compounds. Nano-$Ni_2P$/HT shows high activity for the aqueous hydrogenation reaction of mono- and disaccharides to sugar alcohols.

The hydrogenation of D-glucose to produce D-sorbitol has been extensively studied, utilizing a variety of heterogeneous catalysts that includes nickel [101–115], cobalt [116–118], rhodium [103], ruthenium [119–132], or platinum [133,134]. Among these catalysts, sponge nickel catalysts are most commonly employed in the industrial production of D-sorbitol. This preference is due to their composition consisting of low-cost and abundant materials. However, sponge nickel catalysts are prone to pyrophoricity and suffer from rapid deactivation, which can be attributed to metal leaching, sintering of the metal, and the degradation of the support [135,136]. Additionally, these catalysts exhibit low catalytic activity, necessitating the use of high pressures of hydrogen gas and elevated temperatures (ranging from 100 °C to 180 °C and from 50 bar to 150 bar, respectively). Therefore, in alignment with the principles of green and sustainable chemistry, it is of paramount importance to develop air-stable, highly active, and recyclable catalysts for D-glucose hydrogenation.

As depicted in Figure 11, under conditions of 20 bar $H_2$ and 100 °C in an aqueous medium, nano-$Ni_2P$/HT efficiently catalyzes the hydrogenation of D-glucose to produce D-sorbitol with a 99% yield [98]. The resulting D-sorbitol, used in food additives, pharmaceuticals, and cosmetic ingredients, can also be produced at room temperature with a 90% yield (Figure 11b). For practical applications, it is essential to assess the catalyst performance at high D-glucose concentrations. In such conditions, the nano-$Ni_2P$/HT catalyst maintains its efficiency, achieving a 92% yield of D-sorbitol in a 50 wt% D-glucose solution, as demonstrated in Figure 11c. In the aqueous hydrogenation of maltose, nano-$Ni_2P$/HT efficiently produces the desired maltitol, which is second only to D-sorbitol in terms of a production volume and demand among sugar alcohols. Despite the α-1,4-glycosidic bonds in maltose being sensitive to acids and heat, the robust activity of nano-$Ni_2P$/HT under

mild conditions enables the selective production of maltitol without compromising these bonds (Figure 11d) [99]. Nano-$Ni_2P$/HT also shows high activity for the hydrogenation of D-xylose, providing D-xylitol which is commonly used as a sweetener in high yield (Figure 11e) [100]. Even a small amount of catalyst is adequate for facilitating the hydrogenation reactions of maltose and D-xylose, with the respective turnover numbers being 153 and 960. These values are over 150 times higher than those of traditional sponge nickel catalysts, highlighting the superior efficiency of the nano-$Ni_2P$/HT catalyst. Furthermore, after the reaction, nano-$Ni_2P$/HT can be easily separated from the reaction mixture by centrifugation and retains high yields in subsequent uses. Additionally, it demonstrates high durability with no leaching of nickel species from HT during the reaction.

**Figure 11.** Selective hydrogenation of sugars using nano-$Ni_2P$/HT in water. (**a**) Hydrogenation of D-glucose under standard conditions. (**b**) Hydrogenation of D-glucose at room temperature. (**c**) Hydrogenation of D-glucose at a concentration of 50 wt%. (**d**) Hydrogenation of maltose with nano-$Ni_2P$/HT (0.6 mol% Ni). (**e**) Hydrogenation of D-xylose with nano-$Ni_2P$/HT (0.10 mol% Ni).

The proposed reaction mechanism for the hydrogenation of D-glucose catalyzed by the synergistically catalytic behavior of nano-$Ni_2P$ and HT is illustrated in Figure 12. Initially, (I) $H_2$ is activated at the Ni–Ni bond sites on the nano-$Ni_2P$ surface. (II) The adsorbed hydrogen species on the nano-$Ni_2P$ surface spill over onto the interface or the surface of HT. Meanwhile, (III) the aldehyde group of D-glucose is activated at the oxygen-deficient sites on the HT surface. (IV) The activated aldehyde group is then reduced by the spilled-over hydrogen species to form D-sorbitol, thus completing a catalytic cycle. The activation of $H_2$ by nano-$Ni_2P$ was confirmed using $H_2$–$D_2$ exchange reaction and $H_2$–TPD, while the activation of the carbonyl moiety by HT was confirmed through IR analysis. Consequently, the notable catalytic efficiency is attributed to the concurrent activation of $H_2$ by nano-$Ni_2P$ and D-glucose by HT, illustrating their synergistic effect.

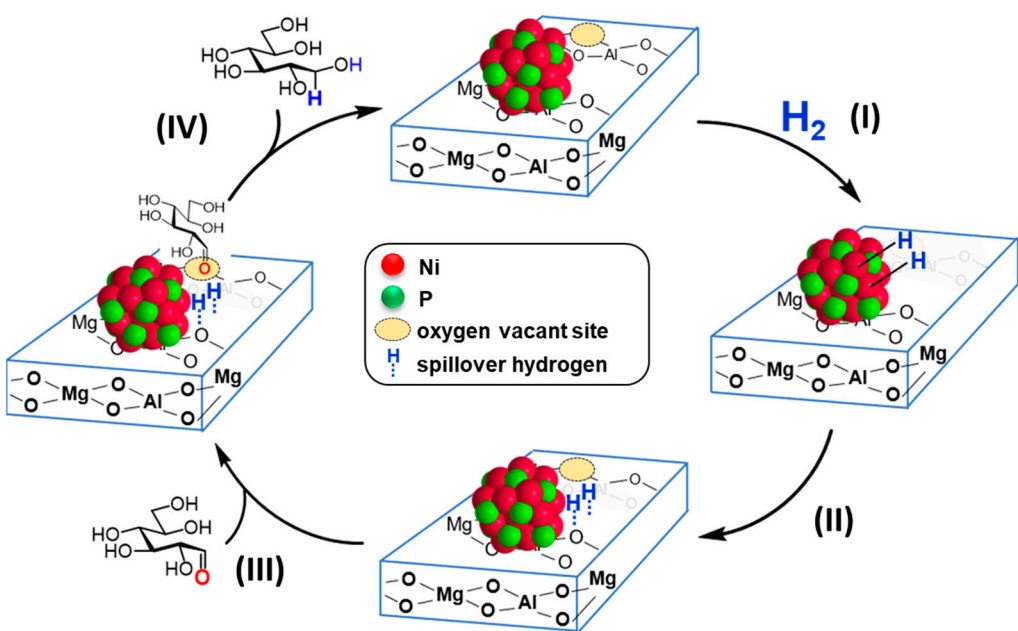

**Figure 12.** A proposed mechanism through the cooperative catalysis of nano-Ni$_2$P with HT. Adapted from [99] Yamaguchi, S.; Fujita, S.; Nakajima, K.; Yamazoe, S.; Yamasaki, J.; Mizugaki, T.; Mitsudome, T. Support-Boosted Nickel Phosphide Nanoalloy Catalysis in the Selective Hydrogenation of Maltose to Maltitol. *ACS Sustain. Chem. Eng.* **2021**, *9*, 6347–6354.

## 3. Conclusions

In this review, we introduce metal phosphide nanoparticle catalysts, particularly cobalt and nickel phosphides, as remarkably efficient heterogeneous catalysts for a variety of reductive molecular transformations. We focused on developing air-stable, non-precious metal catalysts with significantly high activity. These nanosized transition metal phosphides represent a significant advancement over traditional air-unstable non-precious metal catalysts. Their air stability not only eliminates the need for strict anaerobic conditions or pre-reduction treatments, but also enables easy integration with support materials under ambient conditions. In combination with suitable support materials, these metal phosphides exhibit enhanced activity and selectivity in a wide range of hydrogenation reactions.

Cobalt phosphide nanoparticle catalysts, prepared via a solvothermal method, have been shown to be effective in the hydrogenation of nitriles to primary amines, a crucial process in the production of various chemicals. These catalysts demonstrate high activity under atmospheric hydrogen pressure and can be easily recovered and reused, highlighting their practical advantages. Moreover, cobalt phosphide catalysts exhibit high activity in the reductive amination of carbonyl compounds, converting various aldehydes and ketones into primary amines under mild conditions. The structural analysis of these catalysts has revealed that the cobalt species in cobalt phosphide nanoparticles are stabilized in a low oxidation state and possess advantageous Co–Co bonds as well as a higher number of coordinatively unsaturated sites, contributing to their air stability and high catalytic activity for hydrogenation.

Nickel phosphide nanoparticle catalysts exhibit similar non-pyrophoric and air-stable properties and, when combined with support materials, they demonstrate a synergistically enhanced catalytic performance. One of the remarkable applications of nickel phosphide nanoparticle catalysts is in the efficient utilization of biomass. For example, nano-Ni$_2$P/HT catalysts show high activity in the aqueous hydrogenation of mono- and disaccharides to sugar alcohols, such as D-sorbitol and maltitol. These sugar alcohols have significant industrial demand and are used in various products such as food additives and pharmaceuticals. Nano-Ni$_2$P/HT maintains high efficiency and selectivity even at high substrate

concentrations and under mild conditions. The proposed reaction mechanism involves the concurrent activation of $H_2$ and the substrate carbonyl moiety by nano-$Ni_2P$ and HT, respectively, demonstrating a synergistic effect.

Non-precious metal phosphide nanoparticle catalysts offer a solution to the limitations of traditional non-precious catalysts by providing high stability, activity, and durability. These attributes make them highly promising for broadening the scope of sustainable molecular transformations and improving current hydrogenation processes. Furthermore, recent advancements have significantly extended the scope of phosphide nanoparticle catalysts. Notably, there has been remarkable progress in the development of iron phosphide catalysts. The iron phosphide catalyst demonstrates high activity, air stability, and durability for the nitrile hydrogenation, while the conventional iron nanoparticle catalysts exhibit no activity under the same conditions [137]. Although this represents a significant step forward in phosphide catalyst technology, it is part of an ongoing journey of discovery and improvement. Additionally, the research has now expanded to include phosphide nanoparticles incorporating precious metals such as ruthenium and palladium. These catalysts have demonstrated remarkable activity and durability, attributed to the ligand effect and ensemble effect [138–140]. They are particularly interesting because of their high sulfur tolerance, which is a significant advantage for certain reductive molecular transformations. These developments open new avenues for the application of metal phosphide catalysts in more diverse and challenging chemical environments. The future of these catalysts is promising, with potential applications in various fields owing to their unique properties and efficiency.

**Funding:** This work was supported by JSPS KAKENHI, grant numbers 20H02523, 21K04776, 23KJ1484, and 23H01761 and JST PRESTO, grant number JPMJPR21Q9. It was partially supported by JST-CREST, grant number JPMJCR21L5, and the Cooperative Research Program of the Institute for Catalysis, Hokkaido University, grant number 21A1005.

**Acknowledgments:** I would like to thank M. Sheng, H. Ishikawa, T. Tsuda, S. Yamaguchi, and T, Mizugaki of Osaka University for their invaluable discussions during the preparation of this review.

**Conflicts of Interest:** The authors declare no conflicts of interest. The funders had no role in the design of the study, in the collection, analyses, or interpretation of data, in the writing of the manuscript, or in the decision to publish the results.

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
