# Peer review of "Air-Stable and Highly Active Transition Metal Phosphide Catalysts for Reductive Molecular Transformations"

_catalysts, doi:10.3390/catal14030193_

Round 1

Reviewer 1 Report

Comments and Suggestions for Authors

A brief but concise review of T. Mitsudome focuses, as the name implies, on air-stable and highly active transition metal phosphide catalysts for reductive molecular transformations. The novelty and relevance of the topic does not need comments. It is recommended to accept and publish the manuscript as soon as possible. Despite the unconditional high scientific significance of the work, there are still some comments that the author can pay attention to.

1.        It seems that the abstract is written too briefly and generally. The reader may get the impression that there is a discussion in the review, for example, of the hydrogenation of LOHC (Liquid organic hydrogen carriers, for example, http://dx.doi.org/10.1016/j.coelec.2022.101207; or https://doi.org/10.1016/j.cej.2023.144836 ), which are not mentioned in the review, but especially since there is Green chemistry in the keywords.

2.        In the field of the topic of the work, it is clear that the author is one of the world's experts. But nevertheless, almost half of the refs are the papers of the author, shouldn't to add to the work of other scientists (for example, https://doi.org/10.1515/psr-2018-0033)?

3.        All figures and tables provided must contain a reference to the source of the citation and permission for publication.

4.        Figures 2,6,10 and Table 3. The values of the fields go into the image and text.

5.        The meaning of bringing Figure 4 to work in this form is not entirely clear. Is it not enough to mention in the text of the manuscript that the yield of this reaction is (93 ± 1)% and practically remains unchanged for up to 7 cycles?

6.        The insert in Figure 8 practically ”lies” on the X axis, which may cause unnecessary questions from readers about the energy values for the spectra shown in the insert.

Reviewer 2 Report

Comments and Suggestions for Authors

This review represents a valuable contribution and compilation of information regarding the use of metal phosphides as catalysts in selective hydrogenation reactions. I find the work to be both interesting and beneficial for the catalytic community dedicated to the hydrogenation of organic compounds. However, for the manuscript to reach its full potential, I recommend considering the following observations:

(1) The addition of a minimum bibliometric study would enhance the manuscript, providing the reader with insights into the current state of the art on the topic. This could include statistics on publications related to metal phosphides as catalysts, the number of existing reviews, and other relevant metrics.

(2) While the manuscript predominantly focuses on cobalt and nickel phosphides, it would be beneficial to include a conceptual diagram in the introduction. This diagram could serve as a guide, outlining the key aspects to be addressed in the review. This may encompass synthesis methods, characterization techniques, properties, and the classification of the studied reactions, providing readers with a clearer roadmap.

(3) Throughout the manuscript and in the conclusions, the authors emphasize the high stability of metal phosphides in air. However, it is suggested to include comparative studies with conventional catalysts for selective hydrogenation. Fig. 4, while informative, may not be sufficient to firmly establish the superior stability of metal phosphides. Additional data or comparisons with traditional catalysts would strengthen this aspect of the manuscript.
